# DISENTANGLING INTER- AND INTRA-VIDEO RELATIONS FOR MULTI-EVENT VIDEO-TEXT RETRIEVAL AND GROUNDING

## ABSTRACT

Video-text retrieval aims to precisely search for videos most relevant to text queries within a video corpus. However, existing methods are largely limited to single-text (single-event) queries and are not effective at handling multi-text (multi-event) queries. Furthermore, these methods typically focus solely on retrieval and do not attempt to locate multiple events within the retrieved videos. To address these limitations, our paper proposes a novel method named Disentangling Inter- and Intra-Video Relations, which jointly addresses multi-event video-text retrieval and grounding. This method leverages both inter-video and intra-video event relationships to enhance retrieval and grounding performance. At the retrieval level, we devise a Relational Event-Centric Video-Text Retrieval module based on the principle that comprehensive textual information leads to precise correspondence between text and video. It incorporates event relationship features at different hierarchical levels and exploits the hierarchical structure of video relationships to achieve multi-level contrastive learning between events and videos. This approach enhances the richness, accuracy, and comprehensiveness of event descriptions, improving alignment precision between text and video and enabling effective differentiation among videos. For event grounding, we propose Event Contrast-Driven Video Grounding, which accounts for positional differences among events on the 2D-temporal Score Map and achieves precise grounding of multiple events through divergence learning for their locations. Our solution not only provides efficient text-to-video retrieval but also accurately grounds events within the retrieved videos, addressing the shortcomings of existing methods. Extensive experimental results on the ActivityNet Captions and Charades-STA benchmark datasets demonstrate the superior performance of our method, validating its effectiveness. The innovation of this research lies in introducing a new joint framework for video-text retrieval and multi-event grounding while offering new ideas for further research and applications in related fields.

## 1 INTRODUCTION

The video-to-text retrieval task (Zhao et al. (2022); Gorti et al. (2022); Liu et al. (2022); Zhang et al. (2023)) aims to accurately identify the video that best matches a text query from a video database. Due to its broad application prospects in video search, recommendation systems, film entertainment, intelligent surveillance, and other fields, it has become a focal point for researchers. Early research primarily focused on single-text (single-event) queries, where a single textual description was used to retrieve the most relevant video. However, given the limited temporal information provided by a single text description, extending this task to multi-text-to-video retrieval, also known as Multi-event Video-Text Retrieval (MVT-R), is essential to address the ambiguity issues associated with single-text descriptions. Compared to single-text queries, multi-text queries offer richer and more detailed event descriptions, enabling the system to better understand the structure and flow of video content, thereby significantly enhancing retrieval performance.

Currently, in the realm of MVT-R, only MeVTR (Zhang et al. (2023)) method has explored matching videos based on multi-text queries. However, the MeVTR method relies on event time labels corresponding to each textual query, significantly increasing annotation costs. Furthermore, MeVTR

Figure 1: Comparison of MVT-R, SVT-RG, and our proposed MVT-RG method: (a) MVT-R retrieves videos relevant to multi-text queries from a video corpus; (b) SVT-RG retrieves videos and grounds specific events based on a single-sentence query; (c) MVT-RG retrieves videos using multi-text queries and grounds the specific events associated with each query.

primarily focuses on retrieval and is unable to perform temporal grounding of multiple events within the retrieved videos. Therefore, achieving video-text retrieval and grounding based on multi-text queries without corresponding time labels for the textual queries presents a substantial challenge. In terms of jointly achieving video retrieval and grounding, JSG (Chen et al. (2023)) proposes a method for Single-event Video-Text Retrieval and Grounding (SVT-RG). This method uses a single sentence as a query to retrieve the corresponding video from a video corpus while also achieving precise temporal grounding. Although effective, this method is specifically designed for single-sentence queries and does not account for the complexity of multi-text queries or the contextual relationships between different events involved in such queries. As a result, JSG still faces significant challenges when dealing with video-text retrieval and grounding based on multi-text queries.

In this paper, we explore a novel task within a weakly supervised setting: Multi-event Video-Text Retrieval and Grounding (MVT-RG). As illustrated in Figure 1, this task is compared with MVT-R and SVT-RG. From this comparison, we observe that the task defined in this paper not only requires the use of multiple texts to retrieve events but also assumes that the event locations in the training samples are unlabeled within the videos. Therefore, MVT-RG requires the efficient extraction of event-related features from each sentence while leveraging the intrinsic connections between texts. This allows us to enable the retrieval of relevant videos from a large corpus and the precise grounding of event timestamps within the videos. To address this challenge, we propose an innovative method, Disentangling Inter- and Intra-Video Relations, which deeply explores the decoupling of inter- and intra-video event relations, covering both retrieval and event grounding.

At the retrieval level, recognizing that more comprehensive textual query information leads to clearer correspondences between texts and videos, we construct the Relational Event-Centric Video-Text Retrieval (RE-CVTR) module. This module aims to decouple relationships between videos, making their features more distinctive. By embedding multi-level event relations into the hierarchical structure of corresponding videos, it enables contrastive learning between events and videos at multiple levels. This enhances the richness, accuracy, and comprehensiveness of event descriptions, improving alignment precision between texts and videos. For event grounding, we observe the positional differences among events on the 2D-temporal Score Map and propose the Event Contrast-Driven Video Grounding (EC-DVG) method. This method successfully separates different events and achieves precise grounding of multiple events. In summary, our approach not only excels in text-to-video retrieval but also accurately locates events in textual queries under weakly supervised settings, addressing the shortcomings of existing methods. In summary, the significant contributions of this research are as follows:

- We introduce a novel task, MVT-RG, and propose the Disentangling Inter- and Intra-Video Relations method. This is the first approach to jointly address multi-text retrieval and event grounding under a weakly supervised setting.

- For video retrieval, recognizing that more comprehensive textual descriptions lead to clearer correspondences with videos, we develop the relational RE-CVTR module. This module embeds features representing event relations at various levels into the hierarchical structure of corresponding video relations, enabling multi-level contrastive learning between events and videos.

- For event grounding, we propose the EC-DVG module. It leverages positional differences of events on the 2D-temporal Score Map and introduces an event position divergence loss,

allowing the model to distinguish multiple events and achieve precise grounding, even without event temporal labels.

## 2 RELATED WORK

**Video-Text Retrieval**. Video-Text Retrieval can be categorized into SVT-R (Single-event Video-Text Retrieval) and MVT-R, based on the number of text queries used. SVT-R focuses on retrieving the video most relevant to a single text query, with the primary challenge being the effective alignment of video and text features. Early works (Torabi et al. (2016); Yu et al. (2018)) in SVT-R concentrated on designing feature alignment mechanisms. With the introduction of the large-scale image-text pre-trained model CLIP (Radford et al. (2021)), many researchers extended (Luo et al. (2022); Zhao et al. (2022)) its application from static images to video. However, the existing methods primarily focused on single-query video retrieval. In contrast, MVT-R requires multiple text queries for video retrieval, a task first introduced in MeVTR. This approach used the CLIP model to extract visual features from videos and text query features from the input queries. It then applied the K-Medoids clustering algorithm to identify key events in the video and calculated the cosine similarity between the CLIP-extracted text query features and the visual features of each key event. Finally, the proposed MeVTR loss function ensured that each text query matched its corresponding event, thereby improving retrieval accuracy. Overall, both SVT-R and MVT-R methods are limited to retrieval and have yet to address the temporal grounding of events within the retrieved videos.

**Video Grounding**. Video Grounding aims to locate specific temporal intervals of events in untrimmed videos based on textual queries. Compared to Video-Text Retrieval, the main challenge in Video Grounding is the precise alignment of visual content with the semantic information in the query. Existing Video Grounding approaches can generally be categorized into supervised and weakly supervised methods. Supervised methods require annotations of event start and end times corresponding to each textual query during training. These methods can be further divided into proposal-based (Zhang et al. (2020a; 2021b); Wang et al. (2021)) and proposal-free (Mun et al. (2020); Zhang et al. (2020b)) approaches, depending on whether a proposal generation module is used. Proposal-based approaches adopt a two-stage model design: first, generating candidate proposals, then jointly modeling these proposals' corresponding video segments and textual queries, and finally selecting the best proposal for the query. A representative work of this type is 2D-TAN (Zhang et al. (2020a)).

Proposal-free approaches, on the other hand, employ a single-stage design, directly predicting event start and end times from the fused video and text features (Zhang et al. (2020b)). Both types have shown impressive performance in Video Grounding. However, these methods assume that the correspondence between videos and textual queries is known during both training and testing. In real-world applications, this correspondence is often unknown, requiring relevant videos to be retrieved from a video corpus before performing the Video Grounding. This adds an extra layer of complexity, as the system must first retrieve the relevant video from a large corpus, then localize the specific event.

To reduce the manual cost of constructing time labels in supervised methods, several studies have focused on Video Grounding under weak supervision. In this setting, only the correspondence between videos and textual queries is provided for model training, without requiring event time labels. Existing weakly supervised Video Grounding(WSVG) methods can be roughly divided into Multiple Instance Learning (MIL)-based (Mithun et al. (2019); Gao et al. (2019)) and reconstruction-based models (Lin et al. (2020); Zheng et al. (2022a); Yang et al. (2021); Zheng et al. (2022b)), depending on their grounding mechanism. Among them, TGA (Mithun et al. (2019)) was the first method to address the weak supervision problem within the MIL framework. It treats videos relevant to the query as positive samples and irrelevant videos as negative samples, achieving fine-grained visual-text alignment by maximizing the matching score of positive samples while minimizing the score for negative ones. Reconstruction-based methods, on the other hand, take videos and textual queries as input and generate proposals that match the queries through inter-modal interactions. The visual features corresponding to these proposals are then used to reconstruct masked textual queries, thereby localizing events. A representative method is SCN (Lin et al. (2020)). Building upon this, LCNet (Yang et al. (2021)) proposed a hierarchical representation of video and text features and introduced self-supervised reconstruction loss to accurately model the local correspondences between video and text, thereby improving WSVGg performance. Meanwhile, CNM (Zheng et al.

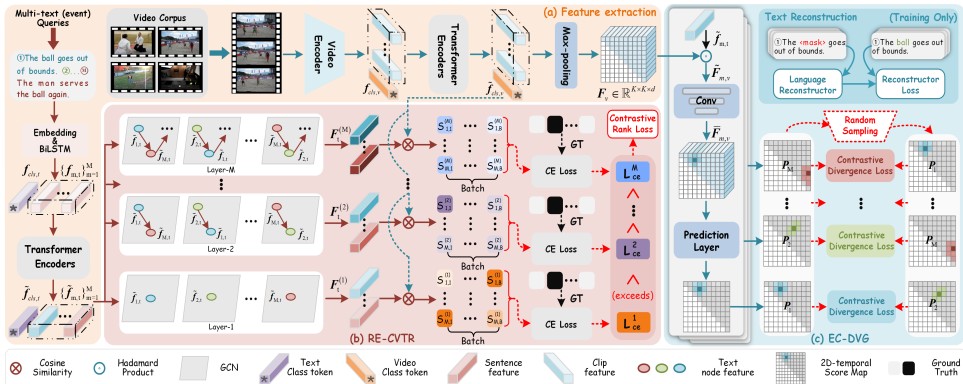

Figure 2: Overview of the Proposed Method: The method comprises three components. (a) presents the feature extraction process, including both video and text feature extraction. (b) illustrates video retrieval, featuring the RE-CVTR. (c) demonstrates EC-DVG. By disentangling inter-video relations in the retrieval module and intra-video relations in the grounding module, the method aligns visual and textual modalities, ensuring accurate correspondence between text and events in the video.

(2022b)) proposed optimizing the feature alignment between video and query through contrastive negative sample mining and reconstruction loss, further enhancing the performance of WSVG tasks. While WSVG methods reduce the manual cost of time label construction, they still assume known correspondence between videos and textual queries during testing, which is not always practical. Recently, JSG proposed SVT-RG. This method aims to retrieve the corresponding video and precisely locate relevant moments using a single sentence (event) as a query. Although JSG performs well in SVT-RG, it struggles with MVT-RG due to its focus on single event.

## 3 METHODOLOGY

### 3.1 PROBLEM STATEMENT AND APPROACH OVERVIEW

Let's first provide a concrete definition for the MVT-RG task. Suppose we have a video corpus consisting of $N$ unedited long videos, denoted as $\boldsymbol{V} = \{\boldsymbol{V}_n\}_{n=1}^N$, where $\boldsymbol{V}_n = \{\boldsymbol{x}_{l,n}\}_{l=1}^L$ and $\boldsymbol{x}_{l,n}$ represents the $l$-th frame of the video, with $L$ being the total number of frames. A multi-text query is represented as $\boldsymbol{Q} = \{\boldsymbol{Q}_m\}_{m=1}^M$, where $\boldsymbol{Q}_m$ denotes the $m$-th textual sub-query, and $M$ is the total number of textual sub-queries for a video. Our goals are twofold: (1) to retrieve the corresponding video $\boldsymbol{V}^*$ from the video corpus $\boldsymbol{V}$ based on the multi-text query $\boldsymbol{Q}$; (2) to locate the time intervals $\boldsymbol{T} = \{(\boldsymbol{t}_{s,m}, \boldsymbol{t}_{e,m})\}_{m=1}^M$ for each textual sub-query $\boldsymbol{Q}_m$ within the retrieved video $\boldsymbol{V}^*$, where $\boldsymbol{t}_{s,m}$ and $\boldsymbol{t}_{e,m}$ represent the start and end times of the event, respectively. It is worth noting that this study adopts a weakly supervised setting. Specifically, during training, we only know the correspondence between each multi-text query and the video, but the specific event time intervals corresponding to each textual sub-query are unknown. During the testing, we have no knowledge of either the correspondence between the multi-text queries and the videos or the specific time intervals of the events described in the text within the videos. The main task of our designed model is to not only retrieve videos that match the textual descriptions from the video corpus but also to accurately locate the time intervals of the events described in the text within the videos.

As illustrated in Figure 2, the method proposed in this paper consists of three components: feature extraction, Relational Event-Centric Video-Text Retrieval (RE-CVTR), and Event Contrast-Driven Video Grounding (EC-DVG). In the feature extraction stage, we meticulously extract features from both videos and texts to ensure an adequate representation of cross-modal information. RE-CVTR innovatively incorporates multi-level relational associations between different events into the modeling framework, employing an event-centric multi-level interaction mechanism to overcome cross-modal challenges in video-text retrieval. For event grounding, EC-DVG introduces an event contrast mechanism and proposes a multi-event grounding strategy based on event differences, enabling more precise event-level cross-modal alignment. Through these innovative designs, the method presented in this paper demonstrates significant performance improvements in simultaneously addressing cross-modal video-text retrieval and multi-event grounding tasks.

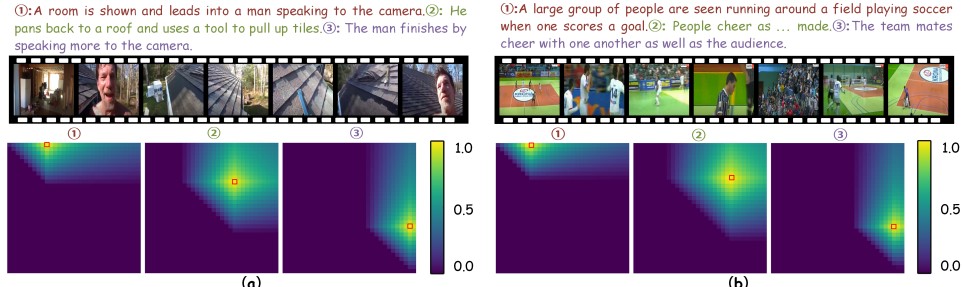

Figure 3: Visualization of the 2D Temporal Score Map with Ground Truth Temporal Labels for Multiple Text Queries on the ActivityNet Captions Dataset. Brighter pixel areas indicate higher probability scores at the current proposal moments. In these Ground Truth score maps, red boxes represent the true locations of the candidate moments corresponding to each sentence.

## 3.2 FEATURE EXTRACTION

**Text Feature Extraction**. Given a multi-text query $\boldsymbol{Q}$ containing $M$ sentences, where each sentence describes an event. Let the $m$-th sentence be denoted as $\boldsymbol{Q}_m = \{q_{m,j}\}_{j=1}^{J}$, where $q_{m,j}$ represents the $j$-th word in the $m$-th sentence, and $J$ is the total number of words in that sentence. We first use Word2Vec (Mikolov et al. (2013)) to tokenize and embed each sentence, converting it into word vectors. The word vectors of the entire sentence are then input into an LSTM, with the final output of the network, $\boldsymbol{f}_{m,t} \in \mathbb{R}^{1 \times d}$, serving as the feature representation of sentence $\boldsymbol{Q}_m$. The feature representations of all sentences in $\boldsymbol{Q}$ can be expressed as $\{\boldsymbol{f}_{1,t}, \boldsymbol{f}_{2,t}, \ldots, \boldsymbol{f}_{M,t}\} \in \mathbb{R}^{M \times d}$. We concatenate the text representations $\{\boldsymbol{f}_{1,t}, \boldsymbol{f}_{2,t}, \ldots, \boldsymbol{f}_{M,t}\}$ with a class token $\boldsymbol{f}_{cls,t} \in \mathbb{R}^{1 \times d}$ to form a new feature representation $\{\boldsymbol{f}_{cls,t}; \boldsymbol{f}_{1,t}, \boldsymbol{f}_{2,t}, \ldots, \boldsymbol{f}_{M,t}\} \in \mathbb{R}^{(M+1) \times d}$. This new feature representation is fed into the text transformer encoder $\boldsymbol{E}_t$, and the output text features are denoted as $\{\tilde{\boldsymbol{f}}_{cls,t}; \tilde{\boldsymbol{f}}_{1,t}, \tilde{\boldsymbol{f}}_{2,t}, \ldots, \tilde{\boldsymbol{f}}_{M,t}\} \in \mathbb{R}^{(M+1) \times d}$.

**Video Feature Extraction:**. For video feature extraction, given an untrimmed video $V$ containing $T$ frames, we first segment the video into $K$ non-overlapping video clips. Then, we use a pre-trained C3D network (Tran et al. (2015)) to extract the visual features of each video clip. The features of the entire video can be represented as $\{\boldsymbol{f}_{1,v}, \boldsymbol{f}_{2,v}, \ldots, \boldsymbol{f}_{K,v}\} \in \mathbb{R}^{K \times d}$. We concatenate these features with a video class token $\boldsymbol{f}_{cls,v} \in \mathbb{R}^{1 \times d}$ to form $\{\boldsymbol{f}_{cls,v}; \boldsymbol{f}_{1,v}, \boldsymbol{f}_{2,v}, \ldots, \boldsymbol{f}_{K,v}\} \in \mathbb{R}^{(K+1) \times d}$. This feature set is fed into the video transformer encoder $\boldsymbol{E}_v$, and the output features are represented as $\{\tilde{\boldsymbol{f}}_{cls,v}; \tilde{\boldsymbol{f}}_{1,v}, \tilde{\boldsymbol{f}}_{2,v}, \ldots, \tilde{\boldsymbol{f}}_{K,v}\} \in \mathbb{R}^{(K+1) \times d}$. Next, we adopt the method from 2D-TAN to construct candidate moments for subsequent event grounding. Specifically, given a candidate moment $T_{ij} = [t_i, t_j]$, where $1 \le i \le j \le K$, we perform max pooling on the sequence features $\tilde{\boldsymbol{f}}_{i,v}, \tilde{\boldsymbol{f}}_{i+1,v}, \ldots, \tilde{\boldsymbol{f}}_{j,v}$ to aggregate them into a single feature vector $\tilde{\boldsymbol{f}}_{ij} \in \mathbb{R}^{1 \times d}$. This aggregated feature $\tilde{\boldsymbol{f}}_{ij}$ serves as the video feature for the candidate moment $T_{ij}$. Finally, we assemble all aggregated features into a temporal feature map $\boldsymbol{F} \in \mathbb{R}^{K \times K \times d}$.

To ensure that the features output by the text and visual branch transformer encoders are highly discriminative, we optimize $\boldsymbol{E}_t$ and $\boldsymbol{E}_v$ using the InfoNCE loss function(He et al. (2020)):

$$\mathcal{L}_{nce} = \mathcal{L}_{nce}^{t2v} + \mathcal{L}_{nce}^{v2t} \tag{1}$$

$$\mathcal{L}_{nce}^{t2v} = -\frac{1}{B} \sum_{b=1}^{B} \left[ \log \frac{e^{S(\tilde{\boldsymbol{f}}_{cls,t}^{b}, \tilde{\boldsymbol{f}}_{cls,v}^{b})\lambda}}{\sum_{z=1}^{B} e^{S(\tilde{\boldsymbol{f}}_{cls,t}^{b}, \tilde{\boldsymbol{f}}_{cls,v}^{z})\lambda}} \right] \tag{2}$$

$$\mathcal{L}_{nce}^{v2t} = -\frac{1}{B} \sum_{b=1}^{B} \left[ \log \frac{e^{S(\tilde{\boldsymbol{f}}_{cls,v}^{b}, \tilde{\boldsymbol{f}}_{cls,t}^{b})\lambda}}{\sum_{z=1}^{B} e^{S(\tilde{\boldsymbol{f}}_{cls,v}^{b}, \tilde{\boldsymbol{f}}_{cls,t}^{z})\lambda}} \right] \tag{3}$$

where $B$ is the batch size, $\lambda$ is a learnable scaling parameter, and $\tilde{\boldsymbol{f}}_{cls,t}^{b}$ and $\tilde{\boldsymbol{f}}_{cls,v}^{b}$ denote the class token features for the $b$-th pair of text and video features within a batch $B$, respectively. $S(\tilde{\boldsymbol{f}}_{cls,t}^{b}, \tilde{\boldsymbol{f}}_{cls,v}^{b})$ represents the cosine similarity between the class token features $\tilde{\boldsymbol{f}}_{cls,t}^{b}$ and $\tilde{\boldsymbol{f}}_{cls,v}^{b}$.

## 3.3 RELATIONAL EVENT-CENTRIC VIDEO-TEXT RETRIEVAL

In the Video-Text Retrieval task, a single text query often corresponds to events that appear in multiple videos. This issue arises from the intrinsic ambiguity of text queries and the repetition of events across various videos, making it more difficult to retrieve the relevant video. To address this, some researchers propose generating multi-text queries for the target video to reduce ambiguity in event descriptions. However, effectively disentangling the relationship between multi-text (event) queries and videos remains challenging. In response, we propose the Relational Event-Centric Video-Text Retrieval (RE-CVTR) module, based on the understanding that more detailed text descriptions facilitate clearer associations between text and video. To fully exploit the hierarchical nature of features that embed event relationships at various levels and their corresponding videos, we propose a hierarchical relationship construction mechanism, as illustrated in Fig. 1(b), enabling multi-level contrastive learning between events and videos.

Specifically, we assume that the input to the RE-CVTR module consists of multi-text query features $\{\tilde{\boldsymbol{f}}_{m,t}\}_{m=1}^{M}$ and video features $\tilde{\boldsymbol{f}}_{cls,v}$. To achieve multi-level interaction and contrastive learning between text queries and videos, we need to create different hierarchies for the multi-text query features, where the input at each hierarchy level is the multi-text query features $\{\tilde{\boldsymbol{f}}_{m,t}\}_{m=1}^{M}$. At the first hierarchical level, the input features $\{\tilde{\boldsymbol{f}}_{m,t}\}_{m=1}^{M}$ are treated as isolated nodes, with no associations constructed between different nodes. This level primarily serves to highlight the individual role of each text. At this level, we concatenate the original text features $\{\tilde{\boldsymbol{f}}_{m,t}\}_{m=1}^{M}$ and denote the result as $\boldsymbol{F}_t^{(1)} = [\tilde{\boldsymbol{f}}_{1,t}, \cdots, \tilde{\boldsymbol{f}}_{M,t}]$. At the second hierarchical level, we treat $\{\tilde{\boldsymbol{f}}_{m,t}\}_{m=1}^{M}$ as root nodes. For each root node feature $\tilde{\boldsymbol{f}}_{m,t}$, we randomly sample a different sentence feature $\tilde{\boldsymbol{f}}_{n,t}$ from the set and feed these two features into a Graph Convolutional Network (GCN) for feature aggregation:

$$\bar{\boldsymbol{f}}_{m,t}^{(2)} = \text{GCN}\left(\left(\tilde{\boldsymbol{f}}_{m,t}, \tilde{\boldsymbol{f}}_{n,t}\right), \boldsymbol{A}_m^{(2)}\right) \quad m, n \in \{1, 2, ..., M\}, m \neq n \tag{4}$$

Here, $\bar{\boldsymbol{f}}_{m,t}^{(2)}$ represents the feature of the $m$-th sentence output by the GCN at the second hierarchy level. $\boldsymbol{A}_m^{(2)} \in \mathbb{R}^{1 \times M}$ is an adjacency matrix consisting of 0s and 1s, which indicate the presence or absence of connections between nodes. Through this approach, we obtain the features at the second hierarchical level, denoted as $\boldsymbol{F}_t^{(2)} = [\bar{\boldsymbol{f}}_{1,t}^{(2)}, \cdots, \bar{\boldsymbol{f}}_{M,t}^{(2)}]$. At each subsequent hierarchical level, we re-randomly sample a new node (sentence) different from the current nodes in the graph and incorporate it into the graph for hierarchical feature aggregation. This process continues until the $M$-th hierarchical level is reached. We denote the text features at all hierarchical levels as $\{\boldsymbol{F}_t^{(l)}\}_{l=1}^{M}$, where $\boldsymbol{F}_t^{(l)} = [\bar{\boldsymbol{f}}_{1,t}^{(l)}, \bar{\boldsymbol{f}}_{2,t}^{(l)}, ..., \bar{\boldsymbol{f}}_{M,t}^{(l)}]$.

To ensure that text query features at each hierarchical level can accurately retrieve corresponding videos, we adopt cross-entropy loss to constrain the retrieval results at each level. This enhances the ability of features at each level to describe events, which is beneficial for subsequent video retrieval and event grounding. Suppose the current batch of video features at the $l$-th level is denoted as $\{\tilde{\boldsymbol{f}}_{cls,v,b}\}_{b=1}^{B}$, where $B$ is the batch size. We compute the cosine similarity between each text feature and each video feature individually. Using this approach, we obtain the cosine similarity score matrix $S^{(l)} \in \mathbb{R}^{M \times B}$ at the $l$-th level:

$$\boldsymbol{S}^{(l)} = \begin{bmatrix} s_{1,1}^{(l)} & s_{1,2}^{(l)} & \cdots & s_{1,B}^{(l)} \\ s_{2,1}^{(l)} & s_{2,2}^{(l)} & \cdots & s_{2,B}^{(l)} \\ \vdots & \vdots & \ddots & \vdots \\ s_{M,1}^{(l)} & s_{M,2}^{(l)} & \cdots & s_{M,B}^{(l)} \end{bmatrix} \tag{5}$$

Here, each element $s_{m,b}^{(l)}$ in $\boldsymbol{S}^{(l)}$ represents the similarity between the $m$-th text query and the $b$-th video features. To compute the cross-entropy loss at each level, we calculate the column-wise average of $\boldsymbol{S}^{(l)}$, resulting in the cosine similarity score vector $\bar{\boldsymbol{S}}^{(l)} = [\bar{s}_1^{(l)}, \bar{s}_2^{(l)}, \ldots, \bar{s}_B^{(l)}]$, where $\bar{\boldsymbol{S}}^{(l)} \in \mathbb{R}^{1 \times B}$. The cross-entropy loss at the $l$-th level is computed as:

$$\mathcal{L}_{ce}^{(l)} = -\sum_{b=1}^{B} y_b \log\left(\bar{s}_b^{(l)}\right) \tag{6}$$

Here, $y_b$ is a label consisting of 0 or 1, where 1 indicates that the text query matches the current video, and 0 indicates no match. The total cross-entropy loss across all levels is expressed as:

$$\mathcal{L}_{ce} = \frac{1}{M} \sum_{l=1}^{M} \mathcal{L}_{ce}^{(l)} \tag{7}$$

In the $M$ hierarchy levels constructed for the multi-text query features $\{\tilde{\boldsymbol{f}}_{m,t}\}_{m=1}^{M}$, each text query at a deeper level acquires more information from other queries within the same video. As we move from the first level to the $M$-th level, the text queries accumulate richer information describing events. This makes the retrieval results at deeper levels more reliable. Based on this, we design a multi-level contrastive ranking loss based on ( Balntas et al. (2016); Zheng et al. (2022a)) to enable the model to perform multi-level contrastive learning. Specifically, we use the cross-entropy loss at each level to evaluate retrieval performance and define the multi-level contrastive ranking loss as follows:

$$\mathcal{L}_{rank} = \frac{1}{(M-1)} \sum_{l=2}^{M} \max \left( \mathcal{L}_{ce}^{(l)} - \mathcal{L}_{ce}^{(l-1)} + h^{*}, 0 \right) \tag{8}$$

Here, $h^{*}$ is a hyperparameter used to control the threshold of the contrastive loss, and $l$ and $l-1$ represent the current level and the previous level, respectively.

## 3.4 EVENT CONTRAST-DRIVEN VIDEO GROUNDING

In Multi-event Video-Text Grounding, multiple textual queries for the same video often describe a series of events that are interrelated yet temporally distinct. These temporal differences lead to different spatial positions for the events on the 2D-temporal Score Map, as illustrated in Figure 3. Manually annotating the timestamps of corresponding events for each query during training would be highly labor-intensive. To address this, we propose the Event Contrast-Driven Video Grounding (EC-DVG) method. This method leverages the positional differences of events on the 2D-temporal Score Map to design a feature learning strategy driven by event position divergence. By utilizing temporal distinctions between events, this strategy effectively differentiates events while preserving their correlations, enabling efficient cross-modal Multi-event Video-Text Grounding without requiring temporal labels. The detailed implementation of EC-DVG is shown in Figure 2(c).

Suppose the feature of the $m$-th sentence query in the multi-text query is denoted as $\tilde{\boldsymbol{f}}_{m,t}$, and the temporal feature map corresponding to the video for this text query is $\boldsymbol{F}_v \in \mathbb{R}^{K \times K \times d}$. We obtain the fused temporal feature map $\tilde{\boldsymbol{F}}_{m,v}$ through Equation (8):

$$\tilde{\boldsymbol{F}}_{m,v}(i,j,:) = \left( \boldsymbol{W}_s \times \tilde{\boldsymbol{f}}_{m,t} \right) \odot \left( \boldsymbol{W}_v \times \boldsymbol{F}_v(i,j,:) \right) \tag{9}$$

where $\boldsymbol{W}_s \in \mathbb{R}^{d \times d}$ and $\boldsymbol{W}_v \in \mathbb{R}^{d \times d}$ are two learnable parameters, and $\odot$ represents the Hadamard product. We then feed $\tilde{\boldsymbol{F}}_{m,v}$ into a convolutional layer to capture the contextual relationships among adjacent candidate temporal features, denoting the output as $\bar{\boldsymbol{F}}_{m,v}$.

To obtain the response scores of different candidate moments for the $m$-th sentence, we feed $\bar{\boldsymbol{F}}_{m,v}$ into a prediction layer consisting of a fully connected layer and a Sigmoid activation function, resulting in a 2D-temporal Score Map $\boldsymbol{P}_m = (p_{m,ij})_{K \times K}$. Here, $p_{m,ij}$ represents the relevance probability score between the candidate moment starting at $i$ and ending at $j$, relative to the $m$-th sentence. By this means, we can generate corresponding 2D-temporal Score Maps for $M$ sentence queries, which we denote as $\boldsymbol{P} \in \mathbb{R}^{K \times K \times M}$, where $\boldsymbol{P} = \{\boldsymbol{P}_m\}_{m=1}^{M}$. To effectively leverage the differences between events and disentangle the correspondences among different textual sub-queries within the video, we propose a Contrastive Loss driven by the divergence of event positions on the 2D-temporal Score Map:

$$\mathcal{L}_{con} = \frac{1}{M} \sum_{m=1}^{M} \| \boldsymbol{P}_m \odot \boldsymbol{P}_n \|_F^2 \quad \text{where} \quad m \neq n \tag{10}$$

Here, $\boldsymbol{P}_n$ is a 2D-temporal Score Map randomly sampled from the set $\boldsymbol{P}$. $\|\cdot\|_F$ denotes the Frobenius norm. By minimizing the loss function in Eq. (10), we encourage events located in $\boldsymbol{P}_m$ and $\boldsymbol{P}_n$

to exhibit spatial dispersion, which preserves the independence of each event's features and disentangles the correlations between events within the video. At the same time, under the constraint of hierarchical relationships, minimizing this loss does not render $\boldsymbol{P}_m$ and $\boldsymbol{P}_n$ completely unrelated. This design ensures that the associations between different events are maintained. Therefore, the loss function in Eq. (10) strikes a good balance between event position diversity and the hierarchical relationships for feature extraction.

## 3.5 Training and Inference

**Training**: The entire network is trained end-to-end, optimizing the model parameters using the following total loss function:

$$\mathcal{L}_{total} = \mathcal{L}_{nce} + \mathcal{L}_{ce} + \mathcal{L}_{rank} + \mathcal{L}_{rec} + \mathcal{L}_{con} \tag{11}$$

where $\mathcal{L}_{rec}$ is the reconstruction loss, which reconstructs the masked text query using the visual features corresponding to the predicted best candidate moment.

**Inference**: During inference, we first compute the cosine similarity between all videos in the video corpus and the text query features at the $M$-th level, selecting the video with the highest similarity as the retrieval result. Subsequently, within the retrieved video, we predict the start and end times of the event corresponding to each sentence query.

## 4 Experiments

### 4.1 Datasets and Evaluation Protocol

**Datasets**. In the domain of video-text retrieval and grounding, the mainstream datasets that suit the task requirements are primarily ActivityNet Captions (Caba Heilbron et al. (2015)) and Charades-STA (Rohrbach et al. (2012a)). Following the protocols established by existing methods, our proposed approach has been evaluated against related methods on these datasets, which encompass videos along with their corresponding multi-text queries. Additionally, to provide a more comprehensive evaluation, we conducted supplementary experiments where our method was further assessed against related approaches on the less commonly used dataset, TaCoS(Regneri et al. (2013)).

**ActivityNet Captions**. The dataset contains 14,926 untrimmed videos, from which 19,811 video-multi-text query pairs are constructed. Notably, a single video may correspond to multiple multi-text queries. The average video duration is 117.60 seconds, with each multi-text query containing an average of 3.63 sentences. The dataset is divided into three subsets: a training set, validation set 1, and validation set 2, containing 10,009, 4,917, and 4,885 video-multi-text query pairs, respectively. Following established protocols, we use validation set 2 as the test set for subsequent evaluations.

**Charades-STA**. The dataset comprises 6,672 videos of indoor activities. We follow the established data split protocol, with 5,338 video-multi-text query pairs in the training set and 1,334 in the test set. On average, the videos are 29.8 seconds long, with each multi-text query corresponding to one video and containing an average of 2.41 sentences.

**Evaluation Protocol**. In this paper, we adopt the same evaluation metrics as the JSG method, namely Recall@K (R@K) and IoU=m, to objectively assess the performance of our model. Specifically, the joint use of R@K and IoU=m indicates the percentage of cases where, for each text query, the model's predicted grounding results in the top K retrieved videos have an Intersection over Union (IoU) exceeding m with the corresponding temporal labels. For ease of comparison with the existing SVT-RG method, the calculation of evaluation metrics in our experiments follows the same approach as methods based on single-sentence queries.

### 4.2 Implementation Details

Similar to the 2D-TAN method, this paper uses a C3D model pre-trained on the UCF101 dataset (Karpathy et al. (2014)) to extract video features. Additionally, word2vec is used to obtain word embeddings. During training, for both the ActivityNet Captions and Charades-STA datasets, we set the batch size to 16 and train the model for 200 epochs, with the hyperparameter $h^*$ set to 0.5.

Table 1: Comparison of the proposed method with state-of-the-art methods on the ActivityNet Captions and Charades-STA datasets. The best and second-best values are highlighted in bold and underline, respectively. JSG** denotes the JSG method adapted for multi-event tasks.

| Method | ActivityNet Captions | | | | | | Charades-STA | | | | | |
| | IoU=0.3 | | IoU=0.5 | | IoU=0.7 | | IoU=0.3 | | IoU=0.5 | | IoU=0.7 | |
| | R@10 | R@100 | R@10 | R@100 | R@10 | R@100 | R@10 | R@100 | R@10 | R@100 | R@10 | R@100 |
| MCN | – | – | 0.18 | 1.26 | 0.09 | 0.70 | – | – | 0.52 | 2.96 | 0.31 | 1.75 |
| CAL | – | – | 0.21 | 1.58 | 0.10 | 0.90 | – | – | 0.75 | 4.39 | 0.42 | 2.78 |
| XML | 3.21 | 12.48 | 1.69 | 7.58 | 0.10 | 0.90 | 0.70 | 2.47 | 0.32 | 1.42 | 0.16 | 0.78 |
| HMAN | – | – | 0.66 | 4.75 | 0.32 | 2.27 | – | – | 1.40 | 7.79 | 1.05 | 4.69 |
| ReLoCLNet | 4.82 | 15.80 | 3.01 | 11.22 | 1.47 | 6.30 | 1.51 | 3.28 | 0.94 | 2.26 | 0.59 | 1.21 |
| MS-SL | 10.80 | 28.31 | 5.85 | 15.65 | 2.46 | 6.60 | 4.46 | 17.61 | 2.55 | 10.05 | 0.91 | 3.76 |
| JSG | 13.27 | 40.61 | 8.76 | 29.98 | 3.83 | 15.78 | 7.23 | 28.71 | 5.67 | 22.50 | 3.28 | 12.34 |
| JSG** | 14.21 | 42.89 | 9.05 | 33.76 | 5.62 | 18.07 | 7.93 | 30.16 | 6.84 | 22.93 | 4.59 | 14.37 |
| **Ours** | **19.54** | **59.45** | **16.58** | **50.37** | **11.13** | **33.45** | **12.82** | **46.25** | **11.62** | **28.54** | **8.92** | **18.83** |

## 4.3 COMPARISON WITH STATE-OF-THE-ART METHODS

In this section, we compare the proposed method with existing approaches on two public datasets, with detailed results shown in Table 1. Additionally, since the MVT-RG task is introduced for the first time in this paper, we only compare our method with the JSG method from SVT-RG to verify its effectiveness. Furthermore, in Table I, we present the results of MCN(Anne Hendricks et al. (2017)), CAL (Escorcia et al. (2019)), XML(Lei et al. (2020)), HMAN(Paul et al. (2021)), ReLoCLNet(Zhang et al. (2021a)), and MS-SL(Dong et al. (2022)), reproduced by the authors of the JSG method, on the ActivityNet Captions and Charades-STA datasets. The original JSG method was designed for single-event video-text retrieval and grounding. We adapted it to create the JSG** version, supporting multi-event tasks for a fair comparison with our method. For details on the adaptation process, please refer to the supplementary materials.

**Results on ActivityNet Captions**. Table 1 presents a performance comparison between the proposed method and currently achievable video retrieval and grounding methods on the ActivityNet Captions dataset. The results demonstrate that our method outperforms existing methods across all evaluation metrics. Specifically, compared to JSG, the proposed method exhibits significant performance improvements in the MVT-RG task, including a 6.27% increase at R@10, IoU=0.3; an 18.84% increase at R@100, IoU=0.3; a 7.82% increase at R@10, IoU=0.5; a 20.39% increase at R@100, IoU=0.5; a 7.30% increase at R@10, IoU=0.7; and a 17.67% increase at R@100, IoU=0.7. Additionally, our method improves by 5.33%, 16.56%, 7.53%, 16.61%, 5.51%,and 15.35% in all evaluation metrics, respectively, compared to JSG**. These notable improvements not only validate the effectiveness of our proposed method but also indicate that MVT-RG has a more pronounced performance advantage over SVT-RG. This is because the proposed method fully leverages the relationships between inter-video and intra-video events, enabling us to achieve more accurate cross-modal matching and grounding.

**Results on Charades-STA**. Table 1 also presents a performance comparison between our method and JSG on the Charades-STA dataset. Notably, the proposed method outperforms existing methods across all evaluation metrics. Specifically, compared to the second-ranked JSG method, our approach achieves improvements of 5.59% at R@10, IoU=0.3; 17.54% at R@100, IoU=0.3; 5.95% at R@10, IoU=0.5; 6.04% at R@100, IoU=0.5; 5.64% at R@10, IoU=0.7; and 6.49% at R@100, IoU=0.7. It is worth noting that our method improves by 4.89%, 16.09%, 4.78%, 5.61%, 4.33%, and 4.46% in all evaluation metrics, respectively, compared to JSG**. These results fully demonstrate the effectiveness and advancement of our method.

**Visualization of Results**. Figure 4 showcases the visualization of retrieval and joint grounding results of our method compared to the JSG method on the ActivityNet Captions dataset. As illustrated in Figure 4 (a), our proposed method achieves correct matching at rank 2 when using multiple text queries. The JSG method focuses on single-event video-text retrieval and grounding. Following the retrieval approach of the JSG method, we apply it independently to each of the three sentence queries, yielding three different retrieval results. Among them, when using the first and third sentences as queries, JSG does not achieve correct matching at rank 10; however, when using the second text query, JSG achieves correct matching at rank 8. This further demonstrates the effectiveness and superiority of our proposed method in the task of multi-event video-text retrieval. Furthermore, the visualization of grounding results in Figure 4 (b) indicates that the event timestamps grounded by

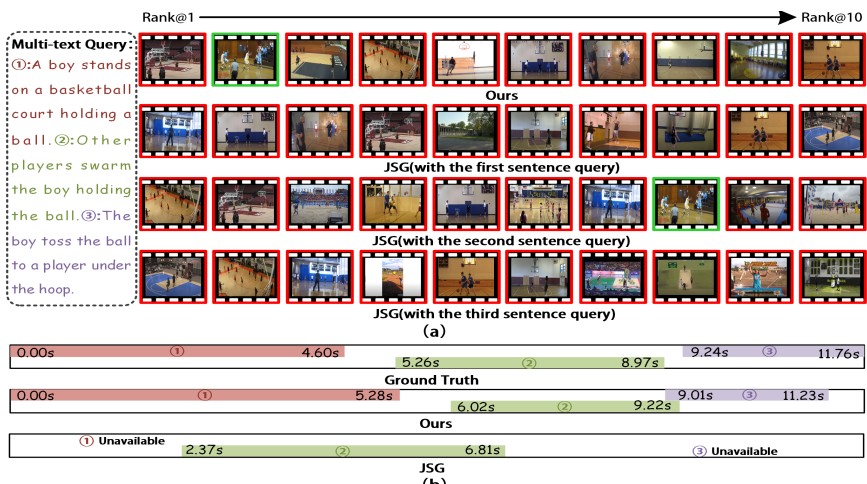

Figure 4: Visualization of Prediction Results for Different Models: (a) shows the top 10 retrieval results from the proposed method in the first row, with the second to fourth rows showing the top 10 results from the JSG model. Green borders indicate correct retrievals, and red borders indicate incorrect ones. (b) presents grounding results based on retrieval outcomes for different models, where "Unavailable" marks a retrieval failure preventing event grounding for the current text query.

our method for each text query are closer to the ground truth, providing additional evidence of the effectiveness of our proposed method in multi-event grounding.

### 4.4 ABLATION STUDIES

The proposed method primarily consists of two modules: RE-CVTR and EC-DVG. To evaluate the individual contributions of these modules, we conduct an ablation study on the ActivityNet Captions dataset, as shown in Table 2. In these experiments, we employ SCN as the baseline model, referred to as "Base", and introduce the InfoNCE loss to fine-tune and optimize it.

**Effectiveness of RE-CVTR**. We integrate RE-CVTR into the baseline model, referred to as 'Base + RE-CVTR.' As shown in Table 3, compared to Base, adding RE-CVTR leads to improvements of 3.95%, 15.02%, 4.01%, 7.05%, 4.15%, and 6.59% across all evaluation metrics. Additionally, to assess the impact of the contrastive ranking loss (CRL), we remove it from the RE-CVTR module, naming it "Base + RE-CVTR w/o CRL". Table 2 demonstrates the efficacy of CRL.

Table 2: Ablation studies on ActivityNet Captions dataset.

| Base | RE-CVTR w/o CRL | RE-CVTR | EC-DVG | IoU=0.3 | | IoU=0.5 | | IoU=0.7 | |
|---|---|---|---|---|---|---|---|---|---|
| | | | | R@10 | R@100 | R@10 | R@100 | R@10 | R@100 |
| ✓ | | | | 13.41 | 40.25 | 10.53 | 36.71 | 4.49 | 20.78 |
| ✓ | ✓ | | | 15.61 | 50.36 | 13.88 | 40.58 | 7.91 | 25.79 |
| ✓ | | ✓ | | 17.36 | 55.27 | 14.54 | 43.76 | 8.64 | 27.37 |
| ✓ | | ✓ | ✓ | 19.54 | 59.45 | 16.58 | 50.37 | 11.13 | 33.45 |

**Effectiveness of EC-DVG.** We integrate EC-DVG into Base+RE-CVTR to evaluate its impact. As shown in Table 2, Base+RE-CVTR+EC-DVG improves by 2.18%, 4.18%, 2.04%, 6.61%, 2.49%, and 6.08% across all metrics, respectively, highlighting EC-DVG's effectiveness in event grounding.

### 5 CONCLUSION

In this paper, we present a novel approach for the MVT-RG task. This method addresses significant limitations in existing techniques that focus on single-text queries and lack the capability to ground multiple events within retrieved videos. By leveraging both inter-video and intra-video event relationships, we enhance retrieval and grounding performance. At the retrieval level, our RE-CVTR module facilitates precise alignment between text and video by utilizing comprehensive textual information and hierarchical event relationships. This multi-level contrastive learning not only enriches event descriptions but also improves alignment precision, allowing effective differentiation among videos. Moreover, the EC-DVG method accounts for positional variations on the 2D temporal Score Map among events, achieving accurate grounding through divergence learning. The experimental results highlight the superior performance of our method compared to existing approaches, validating its effectiveness in addressing the MVT-RG task.

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

# A APPENDIX

## A.1 ADDITIONAL EXPERIMENTS

**TaCoS**. This dataset combines the MPII corpus (Rohrbach et al. (2012b)) with kitchen scene videos, resulting in 127 videos focused on cooking activities. Each video is associated with multiple textual queries. The training set, validation set, and test set contain 1,107, 418, and 380 video-multi-text query pairs, respectively. The average video length in the dataset is 4.79 minutes, and each multi-text query consists of an average of 8.75 sentences.

Table 3: Compare the proposed method with state-of-the-art method on the TaCoS dataset. The bolded data represents the optimal result. The underscore indicates the second-best result. JSG* represents our retrained JSG model for the single-event task on the TaCoS dataset. JSG** denotes the JSG method adapted for multi-event tasks.

| Methods | IoU=0.3 | | IoU=0.5 | | IoU=0.7 | |
|---|---|---|---|---|---|---|
| | R@10 | R@100 | R@10 | R@100 | R@10 | R@100 |
| JSG* | 7.23 | 28.71 | 5.67 | 22.50 | 3.28 | 12.34 |
| JSG** | 8.37 | 31.24 | 7.38 | 25.92 | 3.79 | 14.05 |
| **Ours** | **20.03** | **42.36** | **11.52** | **27.39** | **6.95** | **16.13** |

**Results on TaCoS** . Table 3 presents a performance comparison between the proposed method and JSG on the TaCoS dataset. Since the events in the TaCoS dataset all occur within the same kitchen scene with minimal variation, this presents a significant challenge for event grounding. Despite these difficulties, our method outperforms JSG across all evaluation metrics. Specifically, the proposed method achieves a 12.80% improvement in R@10 with IoU=0.3, a 13.65% improvement in R@100 with IoU=0.3, a 5.85% improvement in R@10 with IoU=0.5, a 4.89% improvement in R@100 with IoU=0.5, a 3.67% improvement in R@10 with IoU=0.7, and a 3.79% improvement in R@100 with IoU=0.7. Since the JSG method is designed for single-event video-text retrieval and grounding, we have adapted it to support multi-event tasks for a fair comparison with our proposed method. Specifically, we modify JSG to calculate cosine similarity scores between each sub-event text query in the multi-event text query and all videos in the video corpus. The scores from each sub-text event query are then summed and averaged, with the video having the highest similarity chosen as the final ranking result for multi-event retrieval. This adaptation allows a fair comparison between JSG and our method under the same framework. As shown in Table 3, even after adapting JSG from handling single-event tasks to multi-event tasks, our method consistently outperforms the adapted JSG** method across all metrics on the TaCoS datasets. These results further confirm the effectiveness and versatility of the proposed method.

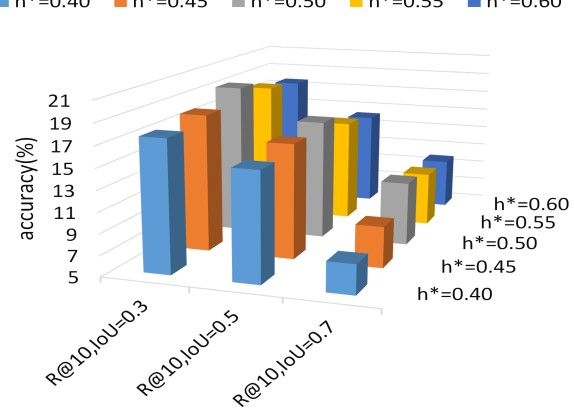

Figure 5: Impact of Different Values of Hyperparameter $h^*$ on Model Performance

## A.2 PARAMETER SELECTION AND ANALYSIS

In Figure 5, we analyze the impact of the hyperparameter $h^*$ in Eq. (8) on the model's performance. This analysis demonstrates how the model's performance varies with different values of $h^*$ on the ActivityNet Captions dataset. The results in Figure 5 show that the model achieves optimal performance when $h^* = 0.5$. Therefore, in all experiments conducted in this paper, we set $h^*$ to 0.5.

## A.3 VISUALIZATION OF MORE RETRIEVAL RESULTS

In Figure 6, we present a visualization of retrieval results and analyze the impact of different ranking strategies on the retrieval subtask. Specifically, the third and fourth columns of Figure 6 illustrate the retrieval results for each sub-event, where cosine similarity scores are calculated between the textual query of each sub-event and all videos in the video corpus. These scores are computed using the first layer and the final (M-th) layer of the RE-CVTR module, respectively. The video with the highest similarity score is selected as the retrieval result for each sub-event. The results demonstrate that retrieval performance using the final (M-th) layer surpasses that of the first layer, validating the effectiveness of the RE-CVTR module, as it leverages hierarchical modeling to extract more comprehensive textual information, leading to more accurate text-video correspondences. Moreover, as shown in the last column of Figure 6, the best retrieval performance is achieved by averaging the cosine similarity scores from the final layer and selecting the video with the highest average score as the retrieval result.

| Video | Multi-text Query | Retrieval Based on First Layer Features $\left(F_i^{(1)}\right)$ | Retrieval Based on the M-th Layer Features $\left(F_i^{(M)}\right)$ | Retrieval Based on the Averaged Cosine Similarity of the M-th Layer |
|---|---|---|---|---|
| | ①:A man is bouncing a tennis ball on a court. | Rank@7 | Rank@2 | |
| | ②:He drops the ball then swings his bat. | Rank@11 | Rank@3 | Rank@1 |
| | ③:He hit it with the racquet over the net, then displays other moves. | Rank@8 | Rank@2 | |
| | ①:A small girl swings on a black swing. | Rank@5 | Rank@3 | |
| | ②:A parent pushes a girl on a black swing. | Rank@10 | Rank@4 | Rank@1 |
| | ③:The girl laughs with glee at being pushed. | Rank@12 | Rank@7 | |
| | ①:Several views are shown of nature and a river. | Rank@8 | Rank@1 | |
| | ②:A large raft filled with people appears. | Rank@15 | Rank@3 | Rank@1 |
| | ③:The people struggle to stay upright as they go through ⋯ falls. | Rank@13 | Rank@1 | |
| | ①:A man is standing in a snow covered parking lot. | Rank@9 | Rank@2 | |
| | ②:He uses a scraper to wipe the snow off the back of a car. | Rank@5 | Rank@1 | Rank@1 |
| | ③:He continues scraping until the windshield is clear. | Rank@17 | Rank@1 | |

Figure 6: Visualization of Retrieval Results with Different Ranking Strategies

