# OpenReview forum: "Disentangling Inter- and Intra-Video Relations for Multi-event Video-Text Retrieval and Grounding"
_ICLR.cc/2025/Conference — Submitted to ICLR 2025_

### Official Review · Reviewer_BzPi · 2024-11-02

**Soundness:** 2
**Presentation:** 3
**Contribution:** 2
**Rating:** 5
**Confidence:** 5

**Summary:**

Considering that existing video-text retrieval methods are largely limited to single-text queries and typically focus solely on retrieval and do not attempt to locate multiple events within the retrieved videos, this paper introduces a new joint framework for video-text retrieval and multi-event grounding, which leverages both inter-video and intra-video event relationships to enhance retrieval and grounding performance. Authors clearly explain the motivation and proposes corresponding methods, and conducts extensive experiments and visualizations to verify them.

**Strengths:**

1.The motivation of this paper is clearly stated, that is, combining MVT-R and SVT-RG, proposing a novel task named MVT-RG and the corresponding approach.

2.The methodology section is well-written and easy to understand. The proposed method can cover the motivation, including the relationship modeling of multiple queries and the implementation of grounding module.

3.The training and inference are detailedly described. Besides, the datasets, implementation details, and evaluation metrics are also clearly explained.

**Weaknesses:**

1.There are some errors that need further verification, including but not limited to MV-TRG on line 186 and Fig. 1 on line 276.

2.Lines 52-53 of the paper claim that MeVTR relies on time labels, but to our knowledge this method does not need time labels.

3.Based on the MVTR task defined in the paper, we do not think that MeVTR belongs to this task. We think that the MVTR task defined is more similar to this work (https://arxiv.org/abs/2201.03639).

4.The experiments in this paper are not complete enough.

(a) The proposed model has both retrieval and grounding capabilities, we hope to see the performance of these two capabilities in the experiment respectively instead of just final results.

(b) The ablation studies only include the ablation of three modules, and there are many details in your approach that need experiments to show their necessity, e.g., each loss in Formula 11 and the necessity of using GCN to model queries relationship.

(c) As far as we know, the accuracy of multi-query retrieval (directly concat multi-query to retrieval) is much better than that of single query (refer to https://arxiv.org/abs/2201.03639). We remain skeptical whether the method in this paper obtains better results because it uses more query information for retrieval. We think that an implementation of JSG based on multi-query retrieval is the suitable baseline of this paper.

(d) The proposed model in this paper is relatively complex. It is recommended to present tests and evaluations of the model's computational overhead.

(e) The paper may also lack the following two aspects of experimentation: visualization of retrieval results and an exploration of the impact of different ranking strategies on the retrieval subtask.

(f) All the SOTA experimental results in this paper are directly taken from the JSG model's experiments on the Event-level Retrieval subtask. However, the JSG model and its methods are designed for single-event query retrieval, may making such a comparison seemingly unfair. The appropriate approach should be to suitably modify the SOTA models to adapt to the novel task proposed in this paper and to detail the modifications within the manuscript. However, the paper does not mention any adaptation of the SOTA models.

**Questions:**

See weaknesses.

---

> ### Author Response · Authors · 2024-11-25
>
> **Weaknesses1**:There are some errors that need further verification, including but not limited to MV-TRG on line 186 and Fig. 1 on line 276.
>
> **Response to W1**: Thank you for your valuable feedback. In the revised version, we will correct this error and thoroughly review the entire manuscript for spelling and clarity.
>
> **Weaknesses2**:Lines 52-53 of the paper claim that MeVTR relies on time labels, but to our knowledge this method does not need time labels.
>
> **Response to W2**: Thank you for pointing out the issue. Upon reviewing the MeVTR[1] method, we noticed that its open-source code does indeed involve the use of time labels for each event. Although this was not explicitly mentioned in the original MeVTR paper, we further contacted the authors of MeVTR and confirmed that time labels are indeed used in the code implementation.
>
> [1] G. Zhang, et al. "Multi-event video-text retrieval."Proceedings of the IEEE/CVF International Conference on Computer Vision (ICCV), 2023, pp. 22113-22123.
>
> **Weaknesses3**:Based on the MVTR task defined in the paper, we do not think that MeVTR belongs to this task. We think that the MVTR task defined is more similar to this work (https://arxiv.org/abs/2201.03639).
>
> **Response to W3**: Thanks for your valuable feedback. Regarding the work you mentioned (https://arxiv.org/abs/2201.03639), we noticed that the multiple text queries used in this work are different descriptions of a single event from various perspectives. For example, the multi-text queries provided in the experimental section are as follows:
> 1. A girl is singing a song.
> 2. A girl is playing the violin.
> 3. A young girl is playing the violin on the beach.
> 4. A small girl standing on sand is playing the violin.
>
> Although these queries differ in their textual descriptions, they actually describe the same event (a girl playing the violin on the beach). Therefore, these queries are not targeting multiple different events in the video, but rather offer different descriptions of the same event.
> In contrast, the MVTR task (multi-event video-text retrieval), first introduced in MeVTR[1], is designed to retrieve corresponding videos using multiple text queries, each describing a different event in the video. The core distinction of the MVTR task is that it handles queries for multiple independent events, rather than different descriptions of the same event as in the work (https://arxiv.org/abs/2201.03639). We hope this explanation helps clarify the difference between the two approaches.

---

> ### Author Response · Authors · 2024-11-25
>
> **Weaknesses4**:The experiments in this paper are not complete enough.
>
> **(a)** The proposed model has both retrieval and grounding capabilities, we hope to see the performance of these two capabilities in the experiment respectively instead of just final results.
>
> **Response to W4(a)**: Thanks for your valuable suggestions. We have provided independent results for both the retrieval and grounding tasks. Specifically, Table I presents a comparison of the proposed method with state-of-the-art methods for the retrieval sub-task. Additionally, JSG** denotes JSG method adapted for multi-event text queries. As shown in Table I, our method consistently outperforms both single-event-based methods and multi-event-based methods (JSG**) across all metrics on the ActivityNet Captions and Charades-STA datasets.
>
> Table II presents a comparison of our method with the JSG [2] and Baseline methods for the grounding sub-task. The evaluation protocol for the grounding task follows the standard settings commonly used in the temporal video grounding domain (R@(1,5), IoU = m). The results demonstrate that our proposed method outperforms both the JSG and Baseline methods across all metrics for the grounding task on the ActivityNet Captions and Charades-STA datasets.
>
> **Table I.  Comparison of the proposed method with state-of-the-art approaches on the ActivityNet Captions and Charades-STA datasets for the retrieval sub-task.** JSG**  **denotes JSG method adapted for multi-event tasks. SumR is the sum of all recall rates. The best values are highlighted in bold.**
>
> | Method    |    |  ActivityNet   |  Captions  |    |    |    |  Charades |  STA  |    |    |
> |:----------|:-----------------------|:-------------|:-------------|:-------------|:-------------|:---------------|:-------------|:-------------|:-------------|:--------------|
> |        | R@1                    | R@5          | R@10         | R@100        | SumR         | R@1            | R@5          | R@10         | R@100        | SumR          |
> | XML       | 5.3                    | 19.4         | 30.6         | 73.1         | 128.4        | 1.6            | 6            | 10.1         | 46.9         | 64.6          |
> | DE++      | 5.3                    | 18.4         | 29.2         | 68           | 121          | 1.7            | 5.6          | 9.6          | 37.1         | 54.1          |
> | ReLoCLNet | 5.7                    | 18.9         | 30           | 72           | 126.6        | 1.2            | 5.4          | 10           | 45.6         | 62.3          |
> | RIVRL     | 5.2                    | 18           | 28.2         | 66.4         | 117.8        | 1.6            | 5.6          | 9.4          | 37.7         | 54.3          |
> | MS-SL     | 7.1                    | 22.5         | 34.7         | 75.8         | 140.1        | 1.8            | 7.1          | 11.8         | 47.7         | 68.4          |
> | JSG       | 6.8                    | 22.7         | 34.8         | 76.1         | 140.5        | 2.4            | 7.7          | 12.8         | 49.8         | 72.7          |
> | JSG**     | 7.3                    | 23.4         | 36.3         | 76.5         | 143.5        | 2.5            | 8.3          | 13.4         | 50.1         | 74.3          |
> | Ours      | **8.4**                    | **26.7**         | **40.1**         | **78.2**         | **153.4**        | **2.9**            | **8.5**          | **14.5**         | **52.7**         | **78.6**         |
>
>
> **Table II. Comparison of the proposed method with JSG and Baseline on the ActivityNet Captions and Charades-STA datasets for the grounding sub-task.** JSG**  **denotes JSG method adapted for multi-event tasks. The best values are highlighted in bold.**
>
> | **Method**   |    |   **ActivityNet Captions** |     |  |     |      |
> |:-------------|:---|:---|:----------------|:-----------------|:----|:-----|
> |              | **R@1, IoU=0.5**                |  **R@1, IoU=0.7**           | **R@5, IoU=0.5**      | **R@5, IoU=0.7**         |
> | JSG      | 24.29                  | 8.26       | 31.45       | 18.43     |
> | Baseline      | 23.71               | 8.87       | 33.62       | 20.12      |
> | Ours     | **27.54**              | **14.33**     | **35.08**    | **24.03**      |
>
>
>
> **Table II. (Continued)**
>
> | **Method**   |    |   **Charades-STA** |     |  |     |      |
> |:-------------|:---|:---|:----------------|:-----------------|:----|:-----|
> |              | **R@1, IoU=0.5**                |  **R@1, IoU=0.7**           | **R@5, IoU=0.5**      | **R@5, IoU=0.7**         |
> | JSG      | 19.31                  | 6.84       | 45.69       | 24.68     |
> | Baseline      | 18.25              | 6.28       | 46.97       | 22.89      |
> | Ours     | **21.57**              | **9.34**     | **50.77**    | **28.26**      |
>
> [2] Z. Chen, et al. "Joint searching and grounding: Multi-granularity video content retrieval." Proceedings of the 31st ACM International Conference on Multimedia, 2023, pp. 975-983.

---

> ### Author Response · Authors · 2024-11-25
>
> **(b)** The ablation studies only include the ablation of three modules, and there are many details in your approach that need experiments to show their necessity, e.g., each loss in Formula 11 and the necessity of using GCN to model queries relationship.
>
> **Response to W4(b)**: Thank you for your valuable feedback. In response to the first concern, we would like to provide the following explanation. In Table 2 of the **revised manuscript**, we present the ablation experiments for the proposed modules and loss functions. Specifically, in the "Base" model, we jointly optimized InfoNCE Loss and Reconstruction Loss during the training phase. For the ablation experiment of the RE-CVTR module, we performed two validations: one is "Base + RE-CVTR," and the other is "Base + RE-CVTR w/o CRL," where we removed the Contrastive Ranking Loss (CRL). Finally, the ablation experiment for the EC-DVG module is based on the "Base + RE-CVTR" model, with the addition of Contrastive Divergence Loss. Through these ablation experiments, we validated the effectiveness of each module and loss function.
>
> Additionally, in response to the second concern, we explored replacing GCN with a "concatenation + convolution" method to model the relationships between queries. In Table III, ① represents the use of the "concatenation + convolution" method, while ② represents the use of the GCN method. As shown in Table III, the GCN-based method significantly outperforms the "concatenation + convolution" method across all metrics on the ActivityNet Captions and Charades-STA datasets.
>
> **Table III Comparison of GCN and “concatenation + convolution” to model the relationships between queries on the ActivityNet Captions and Charades-STA datasets. ① represents the use of the "concatenation + convolution" method, while ② represents the use of the GCN method. The best  values are highlighted in bold .**
> | Method   |    |    |  ActivityNet Captions     | |     |      |
> |:---------|:---|:---|:----------------|:---------|:----|:-----|
> |          | R@10,IoU=0.3                |  R@100,IoU=0.3           | R@10,IoU=0.5      | R@100,IoU=0.5         | R@10,IoU=0.7      | R@100,IoU=0.7
> | ①    | 18.71                  | 56.38        | 15.12         | 49.53        | 10.46         | 32.64        |
> | ②    | **19.54**              | **59.45**     | **16.58**    | **50.37**      | **11.13**    | **33.45**      |
>
> **Table III.（Continued）**
> | Method   |    |    |   Charades-STA|    |    |    |
> |:---------|:---|:---|:---------|:--------|:---|:---|
> |          |R@10,IoU=0.3        |  R@100,IoU=0.3          | R@10,IoU=0.5      | R@100,IoU=0.5           | R@10,IoU=0.7       | R@100,IoU=0.7           |
> | ①    | 12.19           | 45.24        | 11.08         | 26.73         | 8.26          | 17.22         |
> | ②     | **12.82**      | **46.25**      | **11.62**    | **28.54**       | **8.92**      | **18.83**       |

---

> > ### Author Response · Authors · 2024-11-25
> >
> > **(c)** As far as we know, the accuracy of multi-query retrieval (directly concat multi-query to retrieval) is much better than that of single query (refer to https://arxiv.org/abs/2201.03639). We remain skeptical whether the method in this paper obtains better results because it uses more query information for retrieval. We think that an implementation of JSG based on multi-query retrieval is the suitable baseline of this paper.
> >
> > **Response to W4(c)**: To address the concerns raised, we have adapted the JSG method [2], originally designed for single-event video-text retrieval and grounding, to support multi-event tasks for a fair comparison with our proposed approach. Specifically, we modified JSG to calculate cosine similarity scores between each sub-event text query in the multi-event text query and all videos in the video corpus. The scores from each sub-text event query are then summed and averaged, with the video having the highest similarity chosen as the final ranking result for multi-event retrieval. This adaptation allowed a fair comparison between JSG and our method under the same framework, with the results presented in Table I. As shown in Table I, even after adapting JSG from handling single-event tasks to multi-event tasks, our method consistently outperforms the adapted JSG method across all metrics on both the ActivityNet Captions and Charades-STA datasets, thereby demonstrating the effectiveness of our proposed approach. Furthermore, the performance of the JSG** method also significantly surpasses the original JSG method across all metrics, further confirming that multi-text queries provide more accurate results than single-text queries.
> >
> > Upon acceptance, we will release the source code for both our proposed method and the adapted JSG implementation.
> >
> > **Table IV. Comparison of the proposed method with JSG on the ActivityNet Captions and Charades-STA datasets.** JSG** **denotes JSG method adapted for multi-event tasks. The best values are highlighted in bold .**
> > | Method   |    |    |  ActivityNet Captions     | |     |      |
> > |:---------|:---|:---|:----------------|:---------|:----|:-----|
> > |          | R@10,IoU=0.3                |  R@100,IoU=0.3           | R@10,IoU=0.5      | R@100,IoU=0.5         | R@10,IoU=0.7      | R@100,IoU=0.7
> > | JSG**    | 14.21                  | 42.89        | 9.05         | 33.76        | 5.62         | 18.07        |
> > | Ours     | **19.54**              | **59.45**     | **16.58**    | **50.37**      | **11.13**    | **33.45**      |
> >
> > **Table IV.（Continued）**
> > | Method   |    |    |   Charades-STA|    |    |    |
> > |:---------|:---|:---|:---------|:--------|:---|:---|
> > |          |R@10,IoU=0.3        |  R@100,IoU=0.3          | R@10,IoU=0.5      | R@100,IoU=0.5           | R@10,IoU=0.7       | R@100,IoU=0.7           |
> > | JSG**    | 7.93           | 30.16        | 6.84         | 22.93         | 4.59          | 14.37         |
> > | Ours     | **12.82**      | **46.25**      | **11.62**    | **28.54**       | **8.92**      | **18.83**       |
> >
> > [2] Z. Chen, et al. "Joint searching and grounding: Multi-granularity video content retrieval." Proceedings of the 31st ACM International Conference on Multimedia, 2023, pp. 975-983.

---

> ### Author Response · Authors · 2024-11-25
>
> **(d)** The proposed model in this paper is relatively complex. It is recommended to present tests and evaluations of the model's computational overhead.
>
> **Response to W4(d)**: Thank you for your valuable suggestion. We measured the number of parameters and computational complexity of both the Base model and the proposed method. Specifically, the Base model has 24.08M parameters and 19.39G FLOPs, while our method has 24.60M parameters and 21.72G FLOPs. Compared to the Base model, our method introduces only 0.52M additional parameters and an increase of 2.33G in FLOPs.. Despite the increase in computational complexity, our method achieves a significant performance improvement on the ActivityNet Captions dataset. Specifically, the accuracy improves by 6.13% on R10 IoU=0.3, 19.20% on R100 IoU=0.3, 6.05% on R10 IoU=0.5, 13.66% on R100 IoU=0.5, 6.64% on R10 IoU=0.7, and 12.67% on R100 IoU=0.7.
>
> **(e)** The paper may also lack the following two aspects of experimentation: visualization of retrieval results and an exploration of the impact of different ranking strategies on the retrieval subtask.
>
> **Response to W4(e)**: Thank you for the reviewer’s suggestion. We have provided additional visualizations of retrieval results and explored the impact of different ranking strategies on the retrieval subtask in the supplementary materials of the **revised manuscript**.
>
> **(f)** All the SOTA experimental results in this paper are directly taken from the JSG model's experiments on the Event-level Retrieval subtask. However, the JSG model and its methods are designed for single-event query retrieval, may making such a comparison seemingly unfair. The appropriate approach should be to suitably modify the SOTA models to adapt to the novel task proposed in this paper and to detail the modifications within the **revised manuscript**. However, the paper does not mention any adaptation of the SOTA models.
>
> **Response to W4(f)**: Thank you for your valuable feedback. we have adapted the JSG method and compared it with our proposed approach. For details, please refer to Response to W4(c). It is also worth noting that methods such as ReLoCLNet [3] and MS-SL [4] are reproduced versions of JSG for single-event video-text retrieval and grounding. However, since the reproduced code for JSG has not been publicly released, we were unable to adapt these methods to the multi-event task. Additionally, we will provide a detailed explanation of the JSG adaptation process and experimental results in the supplementary materials of the revised version.
>
> [3] M. Zhang, et al. "Multi-stage aggregated transformer network for temporal language localization in videos." Proceedings of the IEEE/CVF Conference on Computer Vision and Pattern Recognition (CVPR), 2021, pp. 12669-12678.\
> [4] J. Dong, et al. "Partially relevant video retrieval." Proceedings of the 30th ACM International Conference on Multimedia, 2022, pp. 246-257.
>
>
> **Thank you for your review and comments. In light of the revisions that we have now made in response to your comments, we kindly request that you reconsider your rating. We are happy to address any further feedback you may have.**

---

> > ### Author Response · Authors · 2024-12-01
> >
> > Dear reviewer BzPi:
> >
> > With the discussion stage ending soon, we wanted to kindly follow up to check if our response has addressed your questions and concerns. If yes, would you kindly consider raising the score before the discussion phase ends? We are truly grateful for your time and effort！
> >
> > Best,
> >
> > Authors of paper 8559.

---

> ### Comment · Reviewer_BzPi · 2024-12-02
>
> After reviewing the authors' rebuttal, I still believe my initial evaluation and score are appropriate. Although the authors have responded to some of the concerns, the main points I raised in my initial review remain unchanged, so I stand by my original judgment.

---

### Official Review · Reviewer_nAgt · 2024-11-04

**Soundness:** 3
**Presentation:** 2
**Contribution:** 2
**Rating:** 5
**Confidence:** 4

**Summary:**

This paper proposes a novel task and a pipeline to address the limitations of single-text (event) queries and effectively handle multi-text (event) queries. The method improves retrieval and grounding performance by combining relationships between videos and internal event relationships. Experimental results on benchmark datasets Charades-STA and ActivityNet Captions demonstrate the effectiveness of the proposed method.

**Strengths:**

1. This paper introduced a novel task setting, Multi-event Video-Text Retrieval. With utmost limited amount of labeling, it is indeed that the challenge of this task should be huge. This setting would make extensive sense in a real world scenario such as online video searching and item recommendation.

2. Event positioning of the two-level grounding can be applied to other tasks.

3. Outperforms current best methods in experiments. The reproducing of the previous methods is informative.

**Weaknesses:**

1. Although the proposed task is highly research-worthy, I don't think the current Video Moment Retrieval (VMR) and Video Corpus Moment Retrieval (VCMR) methods are able to tackle such a difficult task. The previous approach such as JSG is still preliminary, it is certainly not credible to conduct comparative experiments by reproducing them in order to verify the effectiveness of the new method. Therefore, the rationality of the proposed method should be re-considered.

2. The comparison is limited to two benchmark datasets, which requires additional experimentation to prove generalizability. Additionally to overview the whole pipeline, I think it may have some error in event localization for certain scenario.

**Questions:**

1. Although the proposed task is highly research-worthy, I don't think the current Video Moment Retrieval (VMR) and Video Corpus Moment Retrieval (VCMR) methods are able to tackle such a difficult task. The previous approach such as JSG is still preliminary, it is certainly not credible to conduct comparative experiments by reproducing them in order to verify the effectiveness of the new method. Therefore, the rationality of the proposed method should be re-considered.

2. The comparison is limited to two benchmark datasets, which requires additional experimentation to prove generalizability. Additionally to overview the whole pipeline, I think it may have some error in event localization for certain scenario.

---

> ### Author Response · Authors · 2024-11-25
>
> **Weaknesses1:** Although the proposed task is highly research-worthy, I don't think the current Video Moment Retrieval (VMR) and Video Corpus Moment Retrieval (VCMR) methods are able to tackle such a difficult task. The previous approach such as JSG is still preliminary, it is certainly not credible to conduct comparative experiments by reproducing them in order to verify the effectiveness of the new method. Therefore, the rationality of the proposed method should be re-considered.
>
>
> **Response to W1:** Thanks for your insightful comments. To address the concerns raised, we have adapted the JSG method [1], originally designed for single-event video-text retrieval and grounding, to support multi-event tasks for a fair comparison with our proposed approach. Specifically, we modified JSG to calculate cosine similarity scores between each sub-event text query in the multi-event text query and all videos in the video corpus. The scores from each sub-text event query are then summed and averaged, with the video having the highest similarity chosen as the final ranking result for multi-event retrieval. This adaptation allowed a fair comparison between JSG and our method under the same framework, with the results presented in Table I. As shown in Table I, even after adapting JSG from handling single-event tasks to multi-event tasks, our method consistently outperforms the adapted JSG method across all metrics on both the ActivityNet Captions and Charades-STA datasets, thereby demonstrating the effectiveness of our proposed approach. Upon acceptance, we will release the source code for both our proposed method and the adapted JSG implementation.
>
>    It is also worth noting that methods such as ReLoCLNet [2] and MS-SL [3] are reproduced versions of JSG [3] for single-event video-text retrieval and grounding. However, since the reproduced code for JSG has not been publicly released, we were unable to adapt these methods to the multi-event task.
>
> **Table I. Comparison of the proposed method with** JSG** **on the ActivityNet Captions and Charades-STA datasets.** JSG** **denotes JSG method adapted for multi-event tasks. The best values are highlighted in** **bold** .
>
> | Method   |    |    |  ActivityNet Captions     | |     |      |
> |:---------|:---|:---|:----------------|:---------|:----|:-----|
> |          | R@10,IoU=0.3                |  R@100,IoU=0.3           | R@10,IoU=0.5      | R@100,IoU=0.5         | R@10,IoU=0.7      | R@100,IoU=0.7
> | JSG**    | 14.21                  | 42.89        | 9.05         | 33.76        | 5.62         | 18.07        |
> | Ours     | **19.54**              | **59.45**     | **16.58**    | **50.37**      | **11.13**    | **33.45**      |
>
> **Table I.（Continued）**
> | Method   |    |    |   Charades-STA|    |    |    |
> |:---------|:---|:---|:---------|:--------|:---|:---|
> |          |R@10,IoU=0.3        |  R@100,IoU=0.3          | R@10,IoU=0.5      | R@100,IoU=0.5           | R@10,IoU=0.7       | R@100,IoU=0.7           |
> | JSG**    | 7.93           | 30.16        | 6.84         | 22.93         | 4.59          | 14.37         |
> | Ours     | **12.82**      | **46.25**      | **11.62**    | **28.54**       | **8.92**      | **18.83**       |
>
> [1] Z. Chen, et al. "Joint searching and grounding: Multi-granularity video content retrieval." Proceedings of the 31st ACM International Conference on Multimedia, 2023, pp. 975-983.\
> [2] M. Zhang, et al. "Multi-stage aggregated transformer network for temporal language localization in videos." Proceedings of the IEEE/CVF Conference on Computer Vision and Pattern Recognition (CVPR), 2021, pp. 12669-12678.\
> [3] J. Dong, et al. "Partially relevant video retrieval." Proceedings of the 30th ACM International Conference on Multimedia, 2022, pp. 246-257.

---

> ### Author Response · Authors · 2024-11-25
>
> **Weaknesses2**:The comparison is limited to two benchmark datasets, which requires additional experimentation to prove generalizability. Additionally to overview the whole pipeline, I think it may have some error in event localization for certain scenario.
>
> **Response to W2:** Thanks for your valuable feedback. The ActivityNet Captions dataset consists of 14,926 untrimmed videos from YouTube, while the Charades-STA dataset includes 6,672 videos depicting indoor activities. In contrast, the TaCoS dataset features kitchen scenes focused on food and cooking. To evaluate the generalizability of our proposed method, we conducted additional experiments on the TaCoS dataset, with the results summarized in Table II. In Table II, JSG* denotes the retrained JSG model for the single-event task on the TaCoS dataset, while JSG** refers to the JSG method adapted for multi-event tasks. As shown in Table II, our proposed method consistently outperforms both JSG* and JSG** across all metrics, further demonstrating its effectiveness and generalizability.
>
> In response to the second concern you raised, "I think it may have some error in event grounding for certain scenarios," we fully understand and highly value this issue. To reduce the cost of constructing time labels for each event, we adopted a weak supervision setup. Specifically, during training, only the correspondence between multi-event text queries and videos is provided, without using temporal labels for individual events. Although the SCN[4] method has shown some effectiveness in decoupling the relationships between different events in a video, it is not sufficient to guarantee accurate event grounding. Notably, each sub-text query in a multi-event text query inherently contains temporal information regarding the start and end times of different events. By transforming this temporal information into 2D coordinates, we observed that these coordinates are distributed across different regions of the 2D temporal score map. Motivated by this observation, we developed the EC-DVG module, which employs contrastive divergence loss to effectively capture the positional differences between events. This approach effectively disentangles events within the video and significantly enhances the precision of event grounding.
>
> Furthermore, in our ablation experiments, the "Base+RE-CVTR+EC-DVG" method, which integrates the EC-DVG module, achieved significant improvements across all evaluation metrics compared to the "Base+RE-CVTR" method. These results highlight the effectiveness of the EC-DVG module in enhancing event grounding accuracy.
>
>
> **Table II. Comparison of the proposed method with** JSG** **on the TaCoS dataset.** JSG** **denotes JSG method adapted for multi-event tasks. The best values are highlighted in** **bold** .
>
> | **Method**   |    |    |      | **TaCoS** |     |      |
> |:-------------|:---|:---|:----------------|:---------|:----|:-----|
> |              | **R@10, IoU=0.3**                |  **R@100, IoU=0.3**           | **R@10, IoU=0.5**      | **R@100, IoU=0.5**         | **R@10, IoU=0.7**      | **R@100, IoU=0.7**           |
> | JSG     | 7.23                  | 28.71       | 5.67       | 22.50        | 3.28        | 12.34        |
> | JSG**      | 8.37                 | 31.24       | 7.38       |25.92        | 3.79         | 14.05      |
> | **Ours**     | **20.03**              | **42.36**     | **11.52**    | **27.39**      | **6.95**    | **16.13**      |
>
>
> [4] Z. Lin, et al. "Weakly-supervised video moment retrieval via semantic completion network." Proceedings of the AAAI Conference on Artificial Intelligence, 2020, 34(07), 11539-11546
>
>
> **Thank you for your review and comments. In light of the revisions that we have now made in response to your comments, we kindly request that you reconsider your rating. We are happy to address any further feedback you may have.**

---

> > ### Author Response · Authors · 2024-12-01
> >
> > Dear reviewer nAgt:
> >
> > With the discussion stage ending soon, we wanted to kindly follow up to check if our response has addressed your questions and concerns. If yes, would you kindly consider raising the score before the discussion phase ends? We are truly grateful for your time and effort！
> >
> > Best,
> >
> > Authors of paper 8559.

---

### Official Review · Reviewer_8vwW · 2024-11-07

**Soundness:** 3
**Presentation:** 3
**Contribution:** 2
**Rating:** 5
**Confidence:** 4

**Summary:**

This paper introduces a novel task, MVT-RG, and propose the Disentangling Inter- and Intra-VideoRelations method. They develop the relational RE-CVTR module and a EC-DVG module to build relationships between video segments. This approach enhances the richness, accuracy, and comprehensiveness of event descriptions, improving alignment precision between text and video and
enabling effective differentiation among videos.

**Strengths:**

1. This paper introduce a extend task MVT-RG which focuses on the multi events localization.
2,They build a novel model containing RE-CVTR module and EC-DVG module.

**Weaknesses:**

1. The experiment metric seems to be different with most of researches for this field.  The wildly used metric is (R@(1,5),IoU = m)
2. The paper uses the 2D map[1] and reconstruction strategy[2] for weakly supervied setting. There are some works based on these paradigms. It is neccessary to introduce them and compare the effectiveness with them.

*[1]W. Yang, T. Zhang, Y. Zhang, and F. Wu, “Local correspondencenetwork for weakly supervised temporal sentence grounding,” IEEE
TIP, vol. 30, 2021
[2]Zheng M, Huang Y, Chen Q, et al. Weakly Supervised Temporal Sentence
Grounding with Gaussian-based Contrastive Proposal Learning [C]. IEEE/CVF Conference on Computer Vision and Pattern Recognition, CVPR 2022, New Orleans, LA, USA,
June 18-24, 2022. 2022: 15534–15543

**Questions:**

1. This paper introduce a novel task,MVT-RG. However, not all of the dataset has the multi event labeling in a same video. How to solve the problem of dataset?

---

> ### Author Response · Authors · 2024-11-25
>
> **Weaknesses1**:The experiment metric seems to be different with most of researches for this field. The wildly used metric is (R@(1,5),IoU = m)
>
> **Response to W1**: Thanks for your insightful comments. In the field of video grounding (moment retrieval or temporal video grounding), the R@(1,5), IoU = m metric is commonly used to measure the grounding accuracy of a model. Specifically, it measures the proportion of the top-K grounding results (K=1 or K=5) whose Intersection over Union (IoU) with the corresponding time label exceeds a predefined threshold m. Different from traditional video grounding, our work addresses video-text retrieval and grounding, a task first proposed by JSG [1]. To ensure a fair comparison with existing methods, we adopt the same evaluation metric as JSG, namely R@K, IoU=m, with consistent settings for K and m. The metric jointly evaluates the model's retrieval and grounding performance. Specifically, for each text query, the model performs grounding on the top-K retrieved videos and calculates the proportion of grounding results whose IoU with the ground-truth temporal annotations exceeds the predefined threshold m.
>
> [1] Z. Chen, et al. "Joint searching and grounding: Multi-granularity video content retrieval." Proceedings of the 31st ACM International Conference on Multimedia, 2023, pp. 975-983.
>
> **Weaknesses2**:The paper uses the 2D map[1] and reconstruction strategy[2] for weakly supervied setting. There are some works based on these paradigms. It is neccessary to introduce them and compare the effectiveness with them.
>
> **Response to W2**: Thanks for your valuable suggestions. In the revised version, we will include references to works [2,3], which are based on these paradigms, in the related work section. However, these methods primarily focus on video grounding tasks, whereas our proposed new task involves multi-event video-text retrieval and grounding. Due to the absence of retrieval functionality in these methods, it is difficult to adapt and transfer them to the framework of our new task, making a fair comparison challenging.
>
> [2] W. Yang, et al. "Local correspondence network for weakly supervised temporal sentence grounding." IEEE Transactions on Image Processing, 30:3252-3262, 2021.\
> [3] M. Zheng, et al. "Weakly supervised temporal sentence grounding with gaussian-based contrastive proposal learning." Proceedings of the IEEE/CVF Conference on Computer Vision and Pattern Recognition (CVPR), 2022, pp.15555-15566.
>
> **Questions1**:This paper introduce a novel task, MVT-RG. However, not all of the dataset has the multi event labeling in a same video. How to solve the problem of dataset?
>
> **Response to Q1**: Thank you for your questions. The MeVTR method focuses on multi-event video-text retrieval tasks. Our work utilizes the ActivityNet Captions and Charades-STA datasets, as employed in the MeVTR[4] method. In both datasets, most videos have corresponding text queries for multiple events, which meet our need for multi-event labels. Additionally, the TaCoS dataset also meets the criteria for multi-event labels. Therefore, we have included comparative experiments on the TaCoS dataset in the supplementary materials.
>
> [4] G. Zhang, et al. "Multi-event video-text retrieval." Proceedings of the IEEE/CVF International Conference on Computer Vision (ICCV), 2023, pp. 22113-22123.
>
> **Thank you for your review and comments. In light of the revisions that we have now made in response to your comments, we kindly request that you reconsider your rating. We are happy to address any further feedback you may have.**

---

> > ### Author Response · Authors · 2024-12-01
> >
> > Dear reviewer 8vwW:
> >
> > With the discussion stage ending soon, we wanted to kindly follow up to check if our response has addressed your questions and concerns. If yes, would you kindly consider raising the score before the discussion phase ends? We are truly grateful for your time and effort！
> >
> > Best,
> >
> > Authors of paper 8559.

---

### Meta-Review · Area_Chair_b2jc · 2024-12-17

**Metareview:**

This paper introduced a novel task setting, Multi-event Video-Text Retrieval. With utmost limited amount of labeling, it is indeed that the challenge of this task should be huge. This setting would make extensive sense in a real world scenario such as online video searching and item recommendation. The reviewers identified that the experimental setting of this paper was inconsistent with previous work, the proposed method was not reasonable, the experimental results were insufficient, and the statements needed to be further verified. The authors' responses did not improve reviewer voting. So the final vote is rejection.

**Additional Comments On Reviewer Discussion:**

See Metareview.

---

### Decision · Program_Chairs · 2025-01-22

Reject